

# The 6-item specific object anthropomorphism scale: a new questionnaire for children and adults

Jonathan David[1,2], Mikaela Stowe[2], Nathan Caruana[3,4] and Melissa M. Norberg[2]

[1] Department of Psychology, York University, Toronto, ON, Canada
[2] School of Psychological Sciences, Macquarie University, Sydney, NSW, Australia
[3] College of Education, Psychology and Social Work, Flinders University of South Australia, Adelaide, SA, Australia
[4] Institute for Mental Health and Wellbeing, Flinders University of South Australia, Adelaide, SA, Australia

## ABSTRACT

The attribution of human characteristics, emotions, or behaviors to nonhuman entities or objects is known as anthropomorphism. Research on anthropomorphism has been limited by abstract measures that may be difficult for both children and adults to understand. To address this issue, we developed and tested the reliability and validity of the Specific Object Anthropomorphism Scale (SOAS) across three studies involving child and adult participants. The SOAS consists of six items that ask respondents to rate the extent to which a specific object possesses anthropomorphic qualities using simple, concrete statements. Study 1 found that the measure fit a one-factor solution in adults (aged 17–72, M = 32.3). In Study 2, we confirmed the unidimensional structure in adults (aged 18–73, M = 32.2) and demonstrated excellent test-retest reliability, convergent validity, and divergent validity. Study 3 confirmed the unidimensional structure in children (aged 5–12, M = 8.3) and showed that the items were highly understandable. Taken together, these findings indicate that the SOAS is a promising measure of anthropomorphic tendencies that may be easier for both children and adults to complete, potentially leading to a better understanding of the nature of anthropomorphism.

## INTRODUCTION

Anthropomorphism has been defined as the capability to attribute humanlike mental, physical, and emotional characteristics to nonhuman agents (*Epley, Waytz & Cacioppo, 2007*). This tendency has been evidenced to varying degrees in both children and adults across cultures, with most adults perceiving slight humanness in inanimate objects, and only a few reporting more extreme perceptions of human qualities (*e.g.*, *Bloom, 2007*; *Chin et al., 2005*; *Epley, Waytz & Cacioppo, 2007*; *Guthrie, 1993*; *Haslam et al., 2008*; *Severson & Lemm, 2016*; *Waytz, Cacioppo & Epley, 2010*). Anthropomorphic tendencies have been linked to variations in socio-emotional functioning (*e.g.*, autistic traits, loneliness, attachment anxiety; *Burgess, Graves & Frost, 2018*; *Caruana, White & Remington, 2021*; *Epley et al.,*

Corresponding author
Jonathan David, jdavid1@yorku.ca

*2008a*; *Kwok et al., 2018*; *White & Remington, 2019*), including how people interact with robots (*Kühne & Peter, 2023*). Anthropomorphism seems more common in children compared to adults and may be adaptive early on when developing socio-emotional skills (*Epley, Waytz & Cacioppo, 2007*), but may be a risk factor for maladaptive consequences in adults (*e.g.*, hoarding; *Neave et al., 2015*; *Norberg et al., 2018*; *Norberg et al., 2020*; *Timpano & Shaw, 2013*). However, the existing research on anthropomorphism has been questioned because it has largely relied on questionnaires which use abstract language that can be difficult for individuals to understand and complete (*Burgess, Graves & Frost, 2018*; *Severson & Lemm, 2016*). This may account for the low endorsement of anthropomorphic tendencies in adults. Given the differential processes across the lifespan, and difficulties in assessing anthropomorphism, we sought to develop a new measure of anthropomorphism that can be used across the lifespan.

Existing measures of anthropomorphism have been criticized for being too complex for some individuals to complete. For example, the *Individual Differences in Anthropomorphism Questionnaire* (*Waytz, Cacioppo & Epley, 2010*) asks about philosophical concepts that many respondents may interpret as unreasonable or irrational rather than metaphorically (*e.g.*, "To what extent does the ocean have consciousness?"; *Burgess, Graves & Frost, 2018*; *Neave et al., 2015*). As a result, this measure may not capture true variability in anthropomorphic tendencies. The *Anthropomorphic Mental State Ratings* (*Epley et al., 2008a*) and *Graves Anthropomorphic Task Scale* (*Burgess, Graves & Frost, 2018*) are also similarly high-level in the language ability required of respondents, though items in these scales are not as irrational. A less complex scale, the *Individual Differences in Anthropomorphism Questionnaire Child Form*, was developed to provide an age-appropriate measure of children's anthropomorphic tendencies (*Severson & Lemm, 2016*). Although the language is mostly age-appropriate, some questions may still be too complex for children (*e.g.*, "Does the ocean know what it is?"). The *Anthropomorphism Questionnaire* (*Neave et al., 2015*) aims to assess adult's current anthropomorphic tendencies as well as their anthropomorphic tendencies during childhood. However, reflecting on childhood experiences as an adult may introduce recall bias, limiting the validity of the measure. Also, the language level in this measure is still high, so it is not appropriate to adapt this measure for administration in young children.

Some measures of anthropomorphism have been interview-based. For example, in order to measure the anthropomorphic tendencies of young children in an age-appropriate manner, *Tahiroglu & Taylor (2019)* used an adapted version of the Berkeley Puppet Interview. This is a validated interview used to elicit self-reports from children between 4.5- and 7-years-old (*Measelle et al., 1998*). Here, one puppet makes an anthropomorphic statement about an object (*e.g.*, "I think computers can have feelings; they can feel pain or cold"), while the other puppet disagrees with the statement. Children respond by pointing to the puppet with whom they agree. Although enjoyable for children, this measure is resource-intensive and not appropriate for adults.

Given the limitations of existing anthropomorphism measures, we aimed to develop a psychometrically sound measure that can be used to assess the anthropomorphic tendencies of both adults and young children. Developing a standardized instrument for both children

and adults would allow for comparisons between age groups and would also allow researchers to investigate anthropomorphic tendencies longitudinally. To develop this measure, we conducted three studies. During Study 1, we first asked experts to comment on the readability and relevance of an initial set of items for anthropomorphism. After refining our items, we then evaluated the measure's factor structure in a large sample of adults. Based on these findings, we revised the measure again and, in Study 2, evaluated its factor structure in another large adult sample and also established convergent and divergent validity, and test-retest reliability. In study 3, we examined the factor structure and test-retest reliability of the measure in a pediatric sample.

## STUDY 1

### Measure

#### Specific Object Anthropomorphism Scale

To develop this measure, we used an iterative approach driven by both theory and data, as described by *Clark & Watson* (*2016*; *2019*). Authors MMN and NC initially wrote a 12-item pool, by first reviewing existing anthropomorphism measures and using this knowledge to develop new items that were at a second-grade reading level to ensure its usability across the lifespan. We also avoided statements that were philosophical in nature and that could be interpreted as far-fetched or unreasonable. We chose to assess the anthropomorphic qualities with reference to a specific object to make the statements concrete. Twenty-three anthropomorphism researchers rated these items on their readability and relevance to the construct and provided suggestions. See Table S1 for our initial item pool with relevance and readability ratings. After incorporating feedback and suggestions, we revised our measure to 13 items. Specifically, we made changes to make statements less far-fetched in general, for example by amending the stem of all statements from "This object can…" to "I feel that this object can…". We also removed three statements (*e.g.*, "This object is alive"), as some experts commented they were too unreasonable. In addition, we simplified the language even further. For example, the statement "This object has opinions" was changed to "I feel that this object has likes and dislikes". Finally, we added four expert-suggested statements (*e.g.*, "I feel that this object can get lonely"). For this study, we used a simple 4-point Likert scale from 1 (*not at all*) to 4 (*a lot*), given that three to four response options are optimal for children (*Alan & Atalay Kabasakal, 2020*). We chose four response options because we wanted to capture a greater range of anthropomorphism levels by including more Likert response options.

### Participants

A total of 546 participants were recruited for this study *via* the Macquarie University undergraduate psychology pool, Amazon MTurk, and the community (flyers placed around campus, and advertisements distributed on social media, the International OCD Foundation website, and to our pool of previous participants from our lab studies). This sample size was determined using rules of thumb for factor analytic studies; 300 is considered 'good' whereas 500 is considered 'very good' (*Comrey & Lee, 1992*). Students were awarded course credit, MTurkers were paid USD10.20 per hour, and community

participants were put in a lottery to win AUD50. We excluded 46 participants because 12 did not complete the survey and 34 failed more than one randomly placed attention check item (three attention check items were included in the study; *i.e.,* "*Please respond strongly true for this item if you are paying attention*"). In the final sample ($N = 500$), age ranged from 17–72 ($M = 32.26$, $SD = 11.94$), 61.40% identified as female, 37.60% identified as male, 0.40% identified as nonbinary, and 0.60% preferred not to report their gender. Most participants identified as Caucasian (70.40%), 13.80% as Asian, 7.60% as a mixed ethnicity, and 8.20% identified as various other ethnicities.

## Procedure

Participants completed the study on Qualtrics as part of a larger online survey battery. They were instructed to look at a picture of a stuffed toy with human-like features (*i.e.,* a crocheted cactus (see Supplementary Materials)) and then complete the SOAS. All participants provided written consent and were debriefed at the end of the survey. Ethics approval for this study was obtained from Macquarie University's Human Research Ethics Committee (Approval no: 201950508775).

## Analytic strategy

We first examined the distributions of each item. Exploratory factor analysis was then conducted using the *lavaan* package (*Rosseel, 2012*) in RStudio (*Posit Team, 2024*). We treated the data as categorical because there were only four response options for each item. To determine the number of factors to extract, we conducted parallel analysis (treating the data as continuous), as this leads to the most accurate estimation in categorical data (*Yang & Xia, 2015*). We then used the robust weighted least squares mean and variance-adjusted estimator (*i.e.,* WLSMV estimator) as this is most appropriate for categorical data. We report the Chi-square ($\chi^2$) test although we do not use it to evaluate model fit because of its sensitivity to sample size. We evaluated model fit with the following indices: the comparative fit index (CFI) and Tucker-Lewis Index (TLI) > 0.95, and standardized root-mean-square residual (SRMR) and root-mean-square error of approximation (RMSEA) < 0.06 (*Hu & Bentler, 1998*).

## Results

### Descriptive statistics

When examining the distributions of all items, we found that 50.60–89.60% of participants responded with the lowest score (1; *not at all*) and that 0.80–9.80% responded with the highest score (4; *a lot*). The mean for all items ranged from 1.15–1.89 and the median for all items was 1. Skewness statistics ranged from 0.73–3.62. Thus, the majority of items were positively skewed. See Table 1 for means, standard deviations, skewness statistics, and percentages of participants responding to each option for all Specific Object Anthropomorphism Scale (SOAS) items.

### Exploratory factor analysis

We conducted exploratory factor analysis with all items. Parallel analysis suggested retaining one factor (first three eigenvalues were 7.88, 1.17, and 0.81, whereas the first three random

**Table 1  Descriptive statistics and exploratory factor analysis for Study 1.**

| | % (not at all) | % (a tiny bit) | % (somewhat) | % (a lot) | M (SD) | Skewness | Factor loading (SE) |
|---|---|---|---|---|---|---|---|
| 1. I feel that this object can be happy or sad. | 50.60 | 19.20 | 20.40 | 9.80 | 1.89 (1.05) | 0.73 | .80 (.03) |
| 2. I feel that this object can be naughty or nice. | 67.80 | 18.00 | 10.40 | 3.80 | 1.50 (0.83) | 1.55 | .81 (.02) |
| 3. I feel that this object can think. | 82.40 | 10.80 | 6.00 | 0.80 | 1.25 (0.60) | 2.44 | .92 (.02) |
| 4. I feel that this object can do things on purpose. | 89.60 | 6.60 | 3.00 | 0.80 | 1.15 (0.49) | 3.62 | .90 (.02) |
| 5. I feel that this object has likes and dislikes. | 80.00 | 12.80 | 5.20 | 2.00 | 1.29 (0.66) | 2.42 | .89 (.02) |
| 6. I feel that this object can be kind or mean. | 77.20 | 13.00 | 7.40 | 2.40 | 1.35 (0.72) | 2.09 | .86 (.02) |
| 7. I feel that this object needs friends. | 67.40 | 18.00 | 7.80 | 6.80 | 1.54 (0.90) | 1.60 | .91 (.01) |
| 8. I feel that it is easy to talk to this object. | 61.80 | 18.60 | 14.00 | 5.60 | 1.63 (0.92) | 1.22 | .78 (.03) |
| 9. I feel that this object knows what happens to it. | 80.00 | 12.20 | 5.80 | 2.00 | 1.30 (0.67) | 2.37 | .89 (.02) |
| 10. I feel that this object can be excited or bored. | 81.40 | 11.80 | 5.20 | 1.60 | 1.27 (0.63) | 2.50 | .92 (.02) |
| 11. I feel that this object can be scared or calm. | 79.40 | 13.60 | 5.40 | 1.60 | 1.29 (0.64) | 2.34 | .94 (.01) |
| 12. I feel that this object knows right from wrong. | 87.80 | 8.00 | 3.00 | 1.20 | 1.18 (0.53) | 3.39 | .89 (.02) |
| 13. I feel that this object can get lonely. | 70.80 | 16.00 | 8.20 | 5.00 | 1.47 (0.85) | 1.74 | .93 (.01) |

**Notes.**

N = 500. Items were rated on a 4-point Likert scale from 1 (*not at all*) to 4 (*a lot*). All standardized factor loadings were significant (*p*'s < .001).

data eigenvalues were 1.27, 1.21, and 1.15). The one-factor solution explained 61% of variance and had excellent model fit for three fit indices ($\chi^2$[65] = 242.53, *p* < .001, CFI = 1.00, TLI = 1.00, SRMR = 0.06, RMSEA = .07 (90% CI [.06–.08])), with high factor standardized loadings on all items (.78–.94). See Table 1 for factor loadings for the one-factor solution.

## STUDY 2

Based on Study 1, we made slight modifications to the rating scale in order to address issues with skewness. Although we expected our scale questions to be less skewed in comparison to previous measures, which often contain unreasonable statements, we found that our findings aligned with previous observations of anthropomorphism that it is not commonly endorsed for most adults (*Epley, Waytz & Cacioppo, 2007*; *Neave et al., 2015*; *Waytz, Cacioppo & Epley, 2010*). Similar to previous anthropomorphism measures, 50–90% of adults in our sample did not endorse the statements in the scale. As such, we decided to reduce the rating scale from four to three options. This is because the middle two options (*a tiny bit* & *somewhat*) were likely not adding useful information to discriminate between participants. Thus, we decided to use a 3-point Likert scale, in the hopes that it would reduce the skewness, and also be easier for children to interpret.

The purpose of this study was to confirm the factor structure of the anthropomorphism measure in adults using a modified rating scale, as well as assess test-retest reliability, convergent validity, and divergent validity. We first hypothesized that the new measure would be correlated with previous measures of anthropomorphism (convergent validity) while being less related to a quality-of-life measure (divergent validity). We chose the quality-of-life measure to assess divergent validity because it theoretically should not be

correlated with anthropomorphism. Second, we hypothesized that scores on the new measure would be stable over a two-week period (test-retest reliability).

## Participants

A total of 481 participants were recruited for this study *via* the Macquarie University undergraduate psychology pool, Positly (which recruits Amazon MTurk users), and the community (flyers placed around campus, and advertisements distributed on social media, the the International OCD Foundation website, and to our pool of previous participants from our lab studies). Students were awarded course credit, Positly participants were paid USD7.80 per hour, and community participants were put in a lottery to win AUD75. We wanted to recruit a similarly large sample of participants as in Study 1, as this would also give us adequate power to assess the validity and reliability of the scale. We excluded 61 participants because 50 did not complete the SOAS, 1 reported that their survey responses were dishonest, and 10 failed more than one randomly placed attention check item (*i.e.,* "*Please respond strongly true for this item if you are paying attention*"). In the final sample ($N = 420$), age ranged from 18–73 ($M = 32.15$, $SD = 12.91$), 58.57% identified as female, 40.24% identified as male, 0.48% identified as nonbinary, and 0.71% preferred not to report their gender. Most participants identified as Caucasian (68.10%), 18.33% as Asian, and 13.57% identified as various other ethnicities.

## Measures
### Specific Object Anthropomorphism Scale

Participants were instructed to rate whether they agreed with 13 sentences on a 3-point Likert scale: 0 (*no*), 1 (*maybe*), and 2 (*yes*). We used these modified labels with the aim of increasing the accessibility of the SOAS for children.

### Convergent Validity: Anthropomorphic Mental State Ratings (*Epley et al., 2008a*)

The Anthropomorphic Mental State Ratings (AMSR) is made up of 5-items and measured the extent to which an object had "a mind of its own", "intentions", "free will", "consciousness", and "experienced emotions". Participants rated each item on a 7-point Likert scale from 1 (*not at all*) to 7 (*very much*), with higher scores reflecting higher levels of anthropomorphism. The psychometric properties of the AMSR have not been formally evaluated, but it has demonstrated good internal consistency, and has been correlated with a measure of loneliness, demonstrating criterion validity (*Epley et al., 2008a*). In the current sample, the AMSR had excellent internal consistency ($\omega = .97$).

### Convergent Validity: Graves Anthropomorphic Task Scale (*Burgess, Graves & Frost, 2018*)

The 15-item scale was based on the three-factor theory of anthropomorphism (*Epley, Waytz & Cacioppo, 2007*) and was constructed for use in hoarding research on adults. Participants rated the extent they endorsed each statement on a 7-point Likert scale from 1 (*not at all*) to 7 (*very much*). Higher scores reflect higher levels of anthropomorphism. The Graves Anthropomorphic Task Scale (GATS) has previously demonstrated good validity

and internal consistency, but test-retest reliability has not been assessed (*Burgess, Graves & Frost, 2018*). In the current sample, the GATS had excellent internal consistency ($\omega = .96$).

***Divergent Validity: Quality of Life, Enjoyment and Satisfaction Questionnaire –Short Form (Endicott et al., 1993; Stevanovic, 2011)***

This 16-item scale measures enjoyment and satisfaction with different life domains such as work, mood, physical health, social/family relationships, and daily functioning. Items are rated on a 5-point Likert scale from 1 (*very poor*) to 5 (*very good*) and then the first 14 items are summed and converted to a percentage. Higher percentages indicate better quality-of-life. The Quality of Life, Enjoyment and Satisfaction Questionnaire–Short Form (Q-LES-Q-SF) has previously demonstrated good concurrent validity, internal consistency, and test-retest reliability over one-week (ICC = .93; *Stevanovic, 2011*). In the current sample, the total score had excellent internal consistency ($\omega = .92$).

## Procedure

Participants completed the Q-LES-Q-SF, AMSR, GATS, and SOAS, as part of a larger online survey battery administered through Qualtrics. To facilitate comparison of the anthropomorphism measures, we asked participants to complete them while looking at the same picture from the previous study. Two weeks after completing the survey, participants were invited to complete the SOAS again. All participants provided written consent and were debriefed at the end of the study. Ethics approval for this study was obtained from Macquarie University's Human Research Ethics Committee (Approval no: 5201950508775). The authors received permission to use the Q-LES-Q-SF, AMSR, and GATS from the copyright holders.

## Analytic strategy

We first examined the distributions of each SOAS item. Confirmatory factor analysis was then conducted using the *lavaan* package (*Rosseel, 2012*) in RStudio (*Posit Team, 2024*). To test the unidimensional structure from the previous study, we used the robust weighted least squares estimator (*i.e.,* WLSMV estimator) as it is appropriate for categorical data. We report the same indices of model fit as the previous study and used the same thresholds to evaluate model fit. Additionally, we examined residual bivariate correlations between all items to make sure they were all below $r = |.10|$ (*Kline, 2016*). This would confirm that all variance in the scale items would primarily be accounted for by one factor.

All other analyses were conducted using IBM SPSS version 26 (IBM Corp., Armonk, NY, USA). We tested internal reliability using McDonald's Omega, and established test-retest reliability with an intraclass correlation coefficient (ICC) with a two-way consistency model. Finally, we tested convergent and divergent validity of the SOAS by running Spearman correlations with all other measures (because SOAS scores were positively skewed).

## Results
### Descriptive statistics

When examining the distributions of all items, we found that 53.33–88.57% participants responded with the lowest score (0; *no*) and that 4.29–24.05% responded with the highest

**Table 2  Descriptive statistics and confirmatory factor analysis for Study 2.**

| | % (no) | % (maybe) | % (yes) | M (SD) | Skewness | Factor loading (SE) |
|---|---|---|---|---|---|---|
| 1. I feel that this object can be happy or sad. | 72.62 | 16.43 | 10.95 | 0.38 (0.68) | 1.50 | .86 (.02) |
| 2. I feel that this object can be naughty or nice. | 84.05 | 10.00 | 5.95 | 0.22 (0.54) | 2.41 | .90 (.02) |
| 3. I feel that this object can think. | 86.19 | 8.57 | 5.24 | 0.19 (0.51) | 2.66 | .94 (.01) |
| 4. I feel that this object can do things on purpose. | 88.57 | 7.14 | 4.29 | 0.16 (0.47) | 3.03 | .94 (.02) |
| 5. I feel that this object has likes and dislikes. | 81.19 | 11.90 | 6.90 | 0.26 (0.57) | 2.13 | .92 (.02) |
| 6. I feel that this object can be kind or mean. | 84.29 | 9.05 | 6.67 | 0.22 (0.55) | 2.40 | .95 (.01) |
| 7. I feel that this object needs friends. | 69.52 | 19.76 | 10.71 | 0.41 (0.68) | 1.37 | .83 (.03) |
| 8. I feel that it is easy to talk to this object. | 53.33 | 22.62 | 24.05 | 0.71 (0.83) | 0.59 | .68 (.05) |
| 9. I feel that this object knows what happens to it. | 80.48 | 11.67 | 7.86 | 0.27 (0.60) | 2.04 | .94 (.01) |
| 10. I feel that this object can be excited or bored. | 85.24 | 10.24 | 4.52 | 0.19 (0.50) | 2.59 | .94 (.01) |
| 11. I feel that this object can be scared or calm. | 84.52 | 11.19 | 4.29 | 0.20 (0.49) | 2.51 | .93 (.01) |
| 12. I feel that this object knows right from wrong. | 87.62 | 6.19 | 6.19 | 0.19 (0.53) | 2.77 | .93 (.02) |
| 13. I feel that this object can get lonely. | 76.43 | 14.05 | 9.52 | 0.33 (0.64) | 1.74 | .89 (.02) |

**Notes.**

$N = 420$. Items were rated on a 3-point Likert scale from 0 (*no*) to 2 (*yes*). All standardized factor loadings were significant ($p$'s $< .001$).

score (2; *yes*). The mean for all items ranged from 1.15–1.89 and the median for all items was 0. Skewness statistics ranged from 0.59–3.03. Thus, the items were still skewed but to a lesser extent compared to the previous study. See Table 2 for means, standard deviations, skewness statistics, and percentages of participants responding to each option for all Specific Object Anthropomorphism Scale (SOAS) items.

### Confirmatory factor analysis

We conducted confirmatory factor analysis with all SOAS items and found that the unidimensional model replicated well, with excellent model fit for all four fit indices ($\chi^2[65] = 98.08$, $p = .005$, CFI $= 1.00$, TLI $= 1.00$, SRMR $= 0.05$, RMSEA $= .04$ (90% CI [.02–.05])). Standardized factor loadings ranged from .68–.95 (see Table 2 for all loadings). All residual correlations were below |.10|, except for correlations between items 3 and 7 ($r = -.12$), 4 and 7 ($r = -.13$), 4 and 8 ($r = -.13$), 7 and 8 ($r = .13$), and 7 and 13 ($r = .11$). Because residual correlations were close to |.10|, we decided to retain all items to readminister in Study 3, just in case psychometric performance would be better in children.

### Scale reliability

The SOAS also demonstrated excellent internal reliability ($\omega = .94$). Two hundred seventy-six participants completed the SOAS at follow up (63.57% retention) on average 15 days later (range $= 13$ –34 days). The SOAS demonstrated good test-retest reliability with ICC $= .78$.

### Validity

See Table 3 for Spearman correlations between the SOAS and all other measures. As expected, the SOAS showed statistically significant large correlations with previous measures of anthropomorphism (AMSR & GATS), thus demonstrating good convergent

**Table 3  Means, standard deviations, and spearman correlations with the SOAS in Study 2.**

|  | *M* | *SD* | *r* |
|---|---|---|---|
| SOAS | 3.73 | 5.87 | – |
| AMSR | 8.06 | 6.80 | .65[***] |
| GATS | 27.91 | 18.06 | .82[***] |
| Q-LES-Q-SF | 70.39 | 14.46 | −.05 |

Notes.

*N* = 420.

AMSR, Anthropomorphic Mental State Ratings; GATS, Graves Anthropomorphic Task Scale; SOAS, Specific Object Anthropomorphism Scale; Q-LES-Q-SF, Quality of Life, Enjoyment and Satisfaction Questionnaire –Short Form.

[***] *p* < .001.

validity. The SOAS showed a non-statistically significant correlation with the quality-of-life measure (Q-LES-Q-SF), thus demonstrating divergent validity.

# STUDY 3

The purpose of this study was to confirm the factor structure of the anthropomorphism measure in children aged 5 to 12 years, as well as to determine the test-retest reliability of the measure. Given that this was our final study to test the SOAS, we planned to remove items with poor psychometric performance, or items that were not understood well by children.

## Participants

A total of 120 children were recruited for this study *via* advertisements on social media, newsletters, and websites related to research, education, community, and children/families. Participants were paid AUD10 for their time. The sample size was determined through practical considerations with available resources (*i.e.,* finances for participant reimbursement) and experimenter availability. The children who participated in the current study were aged between 5–12 ($M = 8.34$, $SD = 2.23$) and were 53.33% female and 46.67% male. The majority of children in the current study were identified by their caregiver as Caucasian (85.83%; 6.67% South Asian; 3.33% Asian; 2.50% African American; 1.67% Other).

## Measure
### Specific Object Anthropomorphism Scale
Child participants were instructed to look at the same picture from the previous study, and the experimenter verbally stated each of the 13 items to the child and asked them to rate whether they agreed with each statement on a 3-point Likert scale: 0 (*no*), 1 (*maybe*), and 2 (*yes*).

## Procedure
The experimenter first obtained written consent from the parent and verbal consent from the child. We verbally administered the SOAS to children in front of their parents. Doing so meant that we could monitor for any comprehension issues. If a child did not understand an item, the experimenter rephrased the item for them. Parents were allowed to interject
if they felt it was needed. All items were understood by child participants, with only two children (1.7%) requiring item 4 ("I feel that this object can do things on purpose") to be rephrased. Note that we removed these two responses to item 4 for the psychometric analyses below.

Two weeks later, the child participants were invited to complete the SOAS again. All adult and child participants were verbally debriefed at the end of the study. Ethics approval for this study was obtained from Macquarie University's Human Research Ethics Committee (Approval no: 52021961725162).

## Analytic strategy

We first examined the distributions of each SOAS item. Confirmatory factor analysis was then conducted using the *lavaan* package (*Rosseel, 2012*) in RStudio (*Posit Team, 2024*). To test the unidimensional structure from the previous study, we used the WLSMV estimator and used the same model fit indices as in the previous study. All other analyses were conducted using IBM SPSS version 26 (IBM Corp., Armonk, NY, USA). We tested internal reliability using McDonald's Omega, and established test-retest reliability with an intraclass correlation coefficient (ICC) with a two-way consistency model.

## Results

### Descriptive statistics

See Table 4 for means, standard deviations, skewness statistics, and percentages of child participants responding to each option for all SOAS items. When examining the distributions of all items, we found that 8.33–50.83% of participants responded with the lowest score (0; *no*) and that 29.17–79.17% responded with the highest score (2; *yes*). Mean scores ranged from 0.78–1.71 and skewness statistics ranged from −1.96–0.45. Most items were not skewed in comparison to the adult sample in Study 2.

### Confirmatory factor analysis and item reduction

We conducted confirmatory factor analysis with all SOAS items and found that the unidimensional model had poor model fit ($\chi^2[65] = 82.97$, $p = .07$, CFI = .99, TLI = 0.99, SRMR = 0.10, RMSEA = .05 (90% CI [<.001–.08])), with a low standardized factor loading for item 8 (.15). There were also several pairs of residual correlations that were greater than |.10|, involving items 1, 2, 3, 8, and 13 ($r$'s = |.11|–|.27|). Given the number of residual correlations, we ran Parallel analysis to see if there was more than one factor. However, we found that Parallel analysis suggested extracting one factor (first three eigenvalues were 5.21, 1.36, and 1.07, whereas the first three random data eigenvalues were 1.57, 1.43, and 1.32). We then reran the CFA after removing all of the overlapping items, and also item 4, given two participants needed it to be rephrased and also because of the residual correlations in Study 2's adult sample. We also removed item 6 because it had poor test-retest reliability.[1] The 6-item CFA had excellent model fit ($\chi^2[9] = 2.10$, $p = .99$, CFI = 1.00, TLI = 1.01, SRMR = 0.03, RMSEA <.001 [90% CI = <.001, <.001]). Standardized factor loadings ranged from .59–.94 (see Table 4 for all loadings). Also, all residual correlations were below |.10|. Thus, we used this scale for all of the following analyses.

[1] When computing test-retest reliability with for the 7-item SOAS (items 5, 6, 7, 9, 10, 11, 12), it demonstrated poor test-retest reliability (ICC = .57). To see if any individual items were unreliable, we computed Spearman correlations for each item across time. We found that item 6 had a nonsignificant correlation across time ($r = .12$, $p = .30$), and all other items had significant correlations across time ($r$ s = .25–.58).

**Table 4  Descriptive statistics and confirmatory factor analysis for Study 3.**

| | % (no) | % (maybe) | % (yes) | M (SD) | Skewness | Initial factor loading (SE) | Final factor loading (SE) |
|---|---|---|---|---|---|---|---|
| 1. I feel that this object can be happy or sad. | 8.33 | 12.50 | 79.17 | 1.71 (0.61) | −1.96 | .71 (.10) | – |
| 2. I feel that this object can be naughty or nice. | 16.67 | 20.83 | 62.50 | 1.46 (0.77) | −1.00 | .69 (.07) | – |
| 3. I feel that this object can think. | 47.50 | 20.00 | 32.50 | 0.85 (0.89) | 0.30 | .66 (.06) | – |
| 4. I feel that this object can do things on purpose. | 50.83 | 18.33 | 29.17 | 0.78 (0.88) | 0.45 | .61 (.07) | – |
| 5. I feel that this object has likes and dislikes. | 35.00 | 18.33 | 46.67 | 1.12 (0.90) | −0.23 | .73 (.06) | .76 (.06) |
| 6. I feel that this object can be kind or mean. | 24.17 | 12.50 | 63.33 | 1.39 (0.85) | −0.85 | .76 (.06) | – |
| 7. I feel that this object needs friends. | 30.00 | 16.67 | 53.33 | 1.23 (0.89) | −0.48 | .79 (.05) | .77 (.06) |
| 8. I feel that it is easy to talk to this object. | 40.83 | 15.83 | 43.33 | 1.03 (0.92) | −0.05 | .15 (.11) | – |
| 9. I feel that this object knows what happens to it. | 42.50 | 20.83 | 36.67 | 0.94 (0.89) | 0.12 | .64 (.07) | .59 (.08) |
| 10. I feel that this object can be excited or bored. | 36.67 | 15.00 | 48.33 | 1.12 (0.92) | −0.24 | .87 (.04) | .87 (.04) |
| 11. I feel that this object can be scared or calm. | 32.50 | 15.00 | 52.50 | 1.20 (0.90) | −0.41 | .91 (.03) | .94 (.03) |
| 12. I feel that this object knows right from wrong. | 41.67 | 20.00 | 38.33 | 0.97 (0.90) | 0.07 | .74 (.06) | .73 (.06) |
| 13. I feel that this object can get lonely. | 29.17 | 24.17 | 46.67 | 1.18 (0.86) | −0.35 | .72 (.06) | – |

**Notes.**
$N = 120$. Items were rated on a 3-point Likert scale from 0 (*no*) to 2 (*yes*). All standardized factor loadings were significant ($p$'s $< .001$), except for item 8 ($p = .15$).

### Scale reliability

In the child sample, the 6-item SOAS demonstrated good internal reliability ($\omega = .84$). Seventy-nine (65.83% retention) children completed the SOAS for a second time approximately 14 days later.[2] The 6-item SOAS demonstrated adequate test-retest reliability (ICC $= .60$).

[2] There were errors in data collection/entry which resulted in Part 1 and Part 2 completion dates not being recorded for all participants.

### Comparison with adult sample 2

For comparison with the adult sample, we reran all analyses from Study 2 with the 6-item SOAS and found comparable results. Specifically, we found excellent factor structure, good convergent and divergent validity, and good test-retest reliability (see Supplementary Materials). The total 6-item SOAS score for children ($M = 6.58$, $SD = 3.99$) was significantly higher compared to adults ($M = 1.52$, $SD = 2.77$), $t$ (153.25) $= −13.02$, $p < .001$.

## DISCUSSION

The present study found evidence that the 6-item SOAS is a psychometrically sound scale to measure the anthropomorphic tendencies towards a non-human object in both adults and primary-school aged children. Results from Studies 1, 2, and 3 demonstrated excellent factor structure and internal consistency for the scale, indicating that all items map onto the same construct. The SOAS additionally showed adequate test-retest reliability for both children and adults, indicating its appropriateness for longitudinal studies. As hypothesized, Study 2 found that the SOAS was related to previous measures of anthropomorphism while being less related to a measure of quality-of-life, demonstrating excellent convergent and divergent validity.

Our findings indicate that the SOAS is a developmentally appropriate measure of anthropomorphism in primary school-aged children and adults. The items in the SOAS were designed to be accessible for people with a second-grade reading ability. We found that all items in the 6-item SOAS were understood by all 120 children. Our measure provides an accessible alternative to previous measures of anthropomorphism that have been criticized for being too complex for both children and adults—due to asking philosophical questions (*Burgess, Graves & Frost, 2018*; *Neave et al., 2015*). This criticism also applies to the *Individual Differences in Anthropomorphism Questionnaire Child Form* (*Severson & Lemm, 2016*), which also contains abstract statements albeit with simpler language (*e.g.*, "*Does a mountain have feelings?*"). In comparison, our new measure represents a marked improvement over previous measures because we established it uses short, concrete statements with excellent readability. Thus, it is well positioned to enable future research to investigate the developmental trajectories and correlates of anthropomorphism.

The differences between anthropomorphism levels in children and adults are consistent with the three-factor theory of anthropomorphism, which predicts that children have higher anthropomorphism levels than adults (*Epley, Waytz & Cacioppo, 2007*). Future research should investigate why some individuals continue to anthropomorphize into adulthood, and what kinds of psychological factors predict this. It may be that unmet interpersonal needs and loneliness drive children to anthropomorphize objects in an attempt to find connection (*Epley et al., 2008a*; *Epley, Waytz & Cacioppo, 2007*), and inanimate objects become their primary way of fulfilling their connectedness as adults. There is evidence that anthropomorphism in adulthood is related to increased loneliness, particularly among people with greater hoarding problems (*Burgess, Graves & Frost, 2018*), and autistic people (*Caruana, White & Remington, 2021*). Thus, anthropomorphism may play a role in the development of these issues. The lower scores observed in adults on our measure are consistent with previous anthropomorphism measures, in which scores reflect low endorsement of anthropomorphism. By measuring anthropomorphism using simple, concrete statements, rather than philosophical statements, we believe our measure is a more appropriate tool to examine individual differences in anthropomorphism, even though most adults may endorse low levels of anthropomorphism.

## Theoretical and practical implications

Our new anthropomorphism measure overcomes critical shortcomings of previous scales, which has important theoretical implications. The items in the SOAS are concrete and focused on an actual object, rather than asking participants to respond complex or abstract statements as in previous dispositional measures (*e.g.*, *Burgess, Graves & Frost, 2018*; *Epley et al., 2008b*; *Neave et al., 2015*; *Severson & Lemm, 2016*). Further, because the SOAS can be administered to children as young as five, it allows future research to investigate anthropomorphism longitudinally and compare individual differences between age groups. Consistent with the three-factor theory of anthropomorphism (*Epley, Waytz & Cacioppo, 2007*), we also found that children exhibited greater anthropomorphism scores compared to adults. This theory states that anthropomorphic tendencies in childhood relate to the development of healthy social and emotional functioning, because attributing human

feelings and thoughts in objects becomes a steppingstone for attributing human feelings and thoughts in other children (*i.e.,* theory of mind). Future research should continue to replicate and extend previous research using our measure to gain a better understanding of anthropomorphic tendencies and how later in life, these tendencies relate to loneliness, object attachment, poor social functioning, and autistic traits. Doing so may reveal why some individuals continue to anthropomorphize well into adulthood, and why an adaptive ability early in life (for development of social-emotional functioning) may become related to maladaptive outcomes later on in life.

Our research also has practical applications for advancing several fields of empirical research where the measurement of anthropomorphism is important. First, our anthropomorphism measure has clear applications in clinical psychology research, for investigating the factors that contribute to the maintenance of conditions such as hoarding disorder. For instance, our measure can be used to test whether the anthropomorphism of possessions makes it harder for them to be discarded. Current research examining this relationship has relied on dispositional measures (*e.g.*, *Burgess, Graves & Frost, 2018*; *Neave et al., 2015*; *Timpano & Shaw, 2013*). However, a much more valid approach would be to examine people's anthropomorphism towards their actual possessions, as afforded by our new measure. The SOAS could also be used to investigate anthropomorphism within autistic individuals, including when this trait emerges and how it is related to development of social skills (*e.g.*, expression) in childhood and adolescence. Doing so would require further psychometric evaluation in these populations.

Another practical implication of this new measure is that it could also offer a useful tool for human—robot interaction research, and by extension, artificial intelligence research. Human—robot interaction research aims to understand the factors that lead to positive interactions between humans and social robots. Anthropomorphism has overwhelmingly stood out as a key variable of interest in this field of research, both as a predictor and outcome in studies attempting to determine the human and robot factors that lead to successful outcomes in the application of social robots (*Cross & Ramsey, 2021*). The SOAS could offer a useful tool for informing which individuals are most likely to benefit from robot-mediated interventions in therapeutic (*Guemghar et al., 2022*) or education settings (*Belpaeme et al., 2018*). The SOAS could also be adapted to measure anthropomorphic traits for nonphysical entities, such as artificial intelligence (AI) programs, particularly given the emerging uses of AI in therapeutic and educational settings. However, this will require additional psychometric evaluation.

## Limitations

The current study adopted a novel approach which involved scale items that concretely tied responses to a specific object. In order to standardize our validation of the measure, we exposed all participants to the same novel human-like object and did not re-administer the measure for any other object. Thus, future research may need to assess the psychometric properties of the SOAS with other types of objects, such as objects which do not have visual human-like features. This can simply be done by asking respondents to look at another target object and rate each statement in the SOAS. Second, we did not validate

the SOAS in a sample of adolescents (*e.g.*, participants aged 13–17). Future research could assess the psychometric properties of the SOAS in this age group. Validating the SOAS in adolescents would help to fill the gap in understanding anthropomorphism in this age group, and would be an important step before conducting longitudinal studies tracking anthropomorphism throughout the lifespan. Third, we did not investigate whether the SOAS was correlated with any measures of hoarding, socio-emotional functioning, or autistic traits in children or adults. Given that previous findings have been inconsistent, particularly for child samples, we did not include these measures to assess criterion validity. Future research should use the SOAS to investigate hoarding, autism, and socio-emotional functioning in children and adults.

## CONCLUSIONS

We developed and validated a new scale, called the Specific Object Anthropomorphism Scale (SOAS), which measures anthropomorphic tendencies towards an object. We also showed that the factor structure of the SOAS was replicable across three studies involving adult and pediatric samples. The SOAS demonstrated good internal consistency, test-retest reliability, and convergent and divergent validity. We hope that our new measure will allow future researchers to further examine the correlates and developmental trajectories of anthropomorphism across the lifespan.

## ACKNOWLEDGEMENTS

The authors would like to thank all the anthropomorphism researchers who contributed to the development of this measure, including Ashley Shaw, Kristi M. Lemm, Keong Yap, Deniz Tahiroglu, Maria Sääksjärvi, and Rachel L. Severson (other contributors requested to remain anonymous).

### Funding

Macquarie University provided funding to support this research. The funders had no role in study design, data collection and analysis, decision to publish, or preparation of the manuscript.

### Grant Disclosures

The following grant information was disclosed by the authors:
Macquarie University.

### Competing Interests

Nathan Caruana is an Academic Editor for PeerJ.

### Author Contributions

- Jonathan David conceived and designed the experiments, performed the experiments, analyzed the data, prepared figures and/or tables, authored or reviewed drafts of the article, and approved the final draft.

- Mikaela Stowe performed the experiments, authored or reviewed drafts of the article, and approved the final draft.
- Nathan Caruana conceived and designed the experiments, authored or reviewed drafts of the article, and approved the final draft.
- Melissa M. Norberg conceived and designed the experiments, authored or reviewed drafts of the article, and approved the final draft.

## Human Ethics

The following information was supplied relating to ethical approvals (*i.e.*, approving body and any reference numbers):

Macquarie University Human Research Ethics Committee.

## Data Availability

The data is available in the Supplemental Files.

## Supplemental Information

Supplemental information for this article can be found online at http://dx.doi.org/10.7717/peerj.20153#supplemental-information.

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
