# Peer review of "The 6-item specific object anthropomorphism scale: a new questionnaire for children and adults"

_PeerJ, doi:10.7717/peerj.20153_

## Round 0.1 · original submission · Major Revisions

Reviewer 1 ·

Basic reporting

"PEER REVIEWER ASSESSMENTS:

OBJECTIVE - Full research articles: is there a clear objective that addresses a testable research question(s) (brief or other article types: is there a clear objective)?
Yes - there is a clear objective

DESIGN - Is the current approach (including controls and analysis protocols) appropriate for the objective?
Yes - the approach is appropriate

EXECUTION - Are the experiments and analyses performed with technical rigor to allow confidence in the results?
No - there are minor issues

STATISTICS - Is the use of statistics in the manuscript appropriate?
No - there are issues with the statistics in the study

INTERPRETATION - Is the current interpretation/discussion of the results reasonable and not overstated?
No - there are minor issues

OVERALL MANUSCRIPT POTENTIAL - Is the current version of this work technically sound? If not, can revisions be made to make the work technically sound?
Probably - with minor revisions

PEER REVIEWER COMMENTS:

GENERAL COMMENTS: In this paper, the authors present the results of three studies aimed to a development and validation of a new psychological assessment tool called the Specific Object Anthropomorphism Scale (SOAS). The researchers highlight the limitations of existing measures of anthropomorphism, which use abstract language difficult for both children and adults to understand. Through three studies with adult and child participants, the authors demonstrate the SOAS's reliability and validity using simpler, concrete language to assess the attribution of human qualities to specific objects. The findings indicate the SOAS is a promising measure for understanding anthropomorphic tendencies across different age groups and can facilitate future research in areas like clinical psychology and human-robot interaction.
Although the manuscript is well-written and well-structured, there is still room for improvement.

REQUESTED REVISIONS:
Although the manuscript is well-written and well-structured, there is still room for improvement. I have outlined my suggestions below.

Introduction
A portion of the final paragraph in the Introduction (lines 138 to 146, from ""To develop this measure..."" to ""...to make statements concrete"") would be more appropriately placed in the Method section of Study 1.

Study 1
Lines 157-8: The statement ""Authors MMN and NC initially wrote a 12-item pool"" gives the impression that the items were created from scratch, whereas the previously described procedure suggests they were adapted from existing scales. It would be helpful to clarify this to avoid confusion.

Line 161: The sentence ""we revised our measure to 13 items"" would benefit from additional detail about how the revision was conducted—for example, which item was added or removed, the rationale behind the change, and whether it was based on expert feedback, pilot testing, or statistical analysis. Clarifying this would enhance the transparency and replicability of the procedure.

Line 170: Please specify the exact rule of thumb used to determine the sample size.

Line 173: Indicate the number of attention check questions included in the survey.

Line 183: Could the results have differed if a different image, such as that of an animal or plant, had been used?

Lines 188–202: The Analytic Strategy section appears unnecessarily detailed.

Line 194: Why were alternative methods for factor retention not used, apart from parallel analysis? Given the results presented, it is likely that multiple approaches would have converged on the same conclusion, which would have strengthened the robustness of the findings.

Line 197: Please spell out the full term for WLSMV (Weighted Least Squares Mean and Variance adjusted), and optionally note that it is equivalent to the Diagonally Weighted Least Squares (DWLS) method.

Line 208: Present Ms and SDs for each item in Table 1.

Line 212 and beyond: Please report the results of the WLSMV extraction and the parallel analysis, e.g. , include observed and expected eigenvalues, the percentage of variance explained, and any relevant comparison criteria. Do the same for the other two studies.

Line 214 and beyond: Please use two decimal places consistently when reporting numerical results.

Line 214 and beyond: Avoid including a leading zero before the decimal point for values that are theoretically constrained between 0 and 1 (e.g., p, r, CFI, TLI, RMSEA), in accordance with APA 7 reporting conventions.

Line 214: For RMSEA, please report the 90% confidence interval, and ensure this is done consistently throughout the manuscript.

Study 2
Line 314: Please provide a formal indicator of the extent of skewness, such as the skewness statistic and/or its corresponding standard error, to better quantify the distribution.

Line 320: On the contrary, it appears that the data were skewed to a greater extent. Please provide a formal test or statistical evidence to support this claim.

Considering the results of the first two studies, the number of items could already be reduced here without losing important information.

Line 338: Use a capital letter for ""Spearman"" in Spearman correlation.

Study 3
Line 358: Use two decimal points.

Discussion
The implications of the presented results for human-robot interaction appear to be somewhat distant and speculative."

Experimental design

Please see above

Validity of the findings

Please see above

Additional comments

Please see above

·

Basic reporting

Overall, I found this paper to be well-written, clear, and concise. It adhered to the professional standards of our field. I was fascinated by the development of a new measure that could assess anthropomorphism in both children and adults. The introduction section of the paper provides solid rationale for the need for such a measure. I found that the background literature provided the reader with the foundational knowledge necessary to understand the existing work and the need for a new measure. The tables provided were formatted properly, easy to interpret, and appropriate in that they added value to the paper. Overall, I found the paper made a novel contribution to our field.

Experimental design

I found the study design was appropriate, and the rationale for the methodology was clear. I appreciated that the authors included multiple studies in the paper and that they tested both children and adults. One group missing was adolescents, so it would be interesting to see when developmentally children/adolescents start behaving and responding as adults do. I recommend that the authors discuss how this measure might work with or generalize to other populations. A more detailed explanation of the decision to use the Likert scale and its evolution over time would be helpful for the reader. Perhaps it was the idea that a 3-point Likert scale is easier for children to use. It was interesting that the team opted to use a cactus that was crocheted. Why just this one item? How would the measure work with other types of items? This was mentioned by the authors, but further discussion would be helpful.

Validity of the findings

I would like to see this replicated and tested with other populations. The analyses seem appropriate. The adults and kids tested can lend a lot to the theoretical reasoning and rationale for this work. Overall, I found the work to be ethically sound while making a novel contribution to the existing literature. I think this measure is valuable as it can be adapted to work with both children and adults.

Additional comments

In summary, the manuscript is well-written and clear. I appreciated the rationale provided for the methodological design. I see value in this work for researchers of various fields and would urge the authors to further discuss or elaborate upon the generalizability of this measure to other populations.

---

## Round 0.2 · accepted · Accept

All comments have been addressed, and the limitations of the work have been properly acknowledged.

·

Basic reporting

I found that the revised manuscript has adhered to the professional standards of our field. Changes have been made in the reporting of statistics and tables to comply fully with APA standards.

Experimental design

Specifications of the methods and rationale for the study design have been added. Additional explanations for the analysis plan have also been provided. My comments were addressed by the authors. Specifically, for a need to validate the measure with adolescents and the changes made to the Likert scale in subsequent studies.

Validity of the findings

No additional comments to report for this section at this time.

Additional comments

I feel all my comments have been addressed, and the limitations of the work have been properly acknowledged. I recommend that the manuscript be published at this time.